# An Enhanced Interaction of Graft and Exogenous SA on Photosynthesis, Phytohormone, and Transcriptome Analysis in Tomato under Salinity Stress

**DOI:** 10.3390/ijms251910799

**Published:** 2024-10-08

**Authors:** Chen Miao, Yongxue Zhang, Jiawei Cui, Hongmei Zhang, Hong Wang, Haijun Jin, Panling Lu, Lizhong He, Qiang Zhou, Jizhu Yu, Xiaotao Ding

**Affiliations:** Shanghai Key Laboratory of Protected Horticultural Technology, Horticulture Research Institute, Shanghai Academy of Agricultural Sciences, Jinqi Road No. 1000, Fengxian District, Shanghai 201403, China; miaochen@saas.sh.cn (C.M.);

**Keywords:** salt stress, phytohormone, grafting, ionic content, osmotic substance, transcriptome analysis

## Abstract

Salt stress can adversely affect global agricultural productivity, necessitating innovative strategies to mitigate its adverse effects on plant growth and yield. This study investigated the effects of exogenous salicylic acid (SA), grafting (G), and their combined application (GSA) on various parameters in tomato plants subjected to salt stress. The analysis focused on growth characteristics, photosynthesis, osmotic stress substances, antioxidant enzyme activity, plant hormones, ion content, and transcriptome profiles. Salt stress severely inhibits the growth of tomato seedlings. However, SA, G, and GSA improved the plant height by 22.5%, 26.5%, and 40.2%; the stem diameter by 11.0%, 26.0%, and 23.7%; the shoot fresh weight by 76.3%, 113.2%, and 247.4%; the root fresh weight by 150.9%, 238.6%, and 286.0%; the shoot dry weight by 53.5%, 65.1%, and 162.8%; the root dry weight by 150.0%, 150.0%, and 166.7%, and photosynthesis by 4.0%, 16.3%, and 32.7%, with GSA presenting the most pronounced positive effect. Regarding the osmotic stress substances, the proline content increased significantly by more than 259.2% in all treatments, with the highest levels in GSA. Under salt stress, the tomato seedlings accumulated high Na^+^ levels; the SA, G, and GSA treatments enhanced the K^+^ and Ca^2+^ absorption while reducing the Na^+^ and Al^3+^ levels, thereby alleviating the ion toxicity. The transcriptome analysis indicated that SA, G, and GSA influenced tomato growth under salt stress by regulating specific signaling pathways, including the phytohormone and MAPK pathways, which were characterized by increased endogenous SA and decreased ABA content. The combined application of grafting and exogenous SA could be a promising strategy for enhancing plant tolerance to salt stress, offering potential solutions for sustainable agriculture in saline environments.

## 1. Introduction

Soil salinization is a major environmental stressor that affects seed germination, plant growth, and agricultural productivity, with serious threats to global ecological security [1,2,3]. Approximately 10% of the world’s arable land is currently affected by salinity, a problem worsened by climate change and human activities, leading to annual economic losses of tens of billions of dollars [4]. Key factors contributing to soil salinization include increased evaporation, sea level rise, and the use of high-salinity groundwater for irrigation, which elevates the Na^+^, Cl^−^, and SO_4_^2−^ levels in the soil [5]. However, the affected lands cannot be abandoned because of the finite nature of arable land resources and significant investments in irrigation and drainage infrastructure [4]. Even with low crop yields, these areas remain valuable. Modern greenhouses equipped with precise climate control and advanced systems, such as ventilation and irrigation water disinfection, can provide stable growth conditions for plants. However, salt stress can still arise in these controlled environments, often because of the high salt concentrations in nutrient solutions from plant metabolism, incorrect nutrient ratios, poor drainage, and over-fertilization. This highlights the need for careful management to mitigate salinity-related stress and to ensure sustainable greenhouse gas production.

Salicylic acid (SA) is a plant phenolic hormone that plays a critical role in various physiological and biochemical processes [6,7,8,9]. It regulates plant signaling pathways, strengthens the defense mechanisms, and mediates the responses to both biotic and abiotic stressors [10]. A recent study has revealed that the accumulation of free proline, a key osmolyte, can be significantly higher when both the SA and salt stress are present compared to the salt stress alone [11]. This suggests that SA influences proline metabolism, contributing to cellular turgor maintenance by increasing the free proline levels in lentil plants, thereby enhancing their resistance to salt stress. Other studies have reported that SA can improve plant growth under various abiotic stress conditions by enhancing photosynthesis, regulating osmotic pressure, promoting compatible osmolyte metabolism, and reducing membrane damage [12,13,14]. These findings have indicated the multifaceted role of SA in enhancing plant resilience to stress, particularly in salt-stressed environments.

Grafting is a viable method to cultivate plants with enhanced resistance to adverse environmental conditions. Rootstocks generally have more robust root systems, which provide increased resilience against both biotic and abiotic stressors. Grafting strengthens the plant’s stress resistance and induces various physiological changes that contribute to enhanced resilience [15,16]. The grafting process creates wounds that induce significant stress, initiating a complex wound-healing response involving differential gene expression and hormone signaling [17,18], which is crucial for the plant recovery and the establishment of a functional graft union [19]. Grafting also improves the resistance to drought and salt stress by enhancing the water use efficiency, increasing the photosynthetic rate, and boosting nutrient acquisition capacity [20,21,22]. However, the effects of grafting on photosynthetic characteristics can vary based on the rootstock genotype, demonstrating the importance of selecting an appropriate rootstock to achieve the desired outcomes, particularly under specific environmental conditions [23]. The selection of the correct rootstock can optimize the benefits of grafting and improve plant resilience to various stressors.

Recent studies have investigated methods to enhance plant salt tolerance using exogenous salicylic acid (SA) and grafting techniques [24,25,26,27,28]. Although both SA application and grafting have individually proven effective in improving plant resilience to salt stress, research on their combined effects is limited. This study aimed to explore the interactions and cumulative benefits of using SA and grafting and to discover innovative strategies for increasing plant salt tolerance. We conducted an experiment to assess the potential of combining grafting with exogenous SA as a strategy to mitigate salt stress in tomato plants. We evaluated various plant growth parameters, leaf photosynthesis, and key physiological factors. Moreover, we examined the osmotic substances, the ionic content, and the hormone levels in the leaves. To gain a deeper insight into the underlying molecular mechanisms, we employed RNA-seq technology to analyze the differentially expressed genes (DEGs) in tomato leaves subjected to different treatments. The primary objective was to determine whether the combination of grafting and exogenous SA was more effective in alleviating the salt stress compared to each treatment individually, as indicated by plant phenotypes. Additionally, we aimed to elucidate the physiological mechanisms through which the combined application of grafting and exogenous SA enhances salt tolerance in tomatoes, including the changes in the growth parameters, physiological indices, osmotic regulators, and ion content. By addressing these objectives, we sought to highlight the potential benefits of this combined treatment and contribute to the development of more resilient tomato plants in salt-stress environments.

## 2. Results

### 2.1. Growth Parameters

The tomato seedlings exposed to the salt stress exhibited inhibited growth of both shoots and roots across all treatments. However, the grafting (G) and the application of exogenous salicylic acid (SA) partially mitigated the growth inhibition caused by the salt stress. Notably, the combination of grafting and exogenous SA (GSA) resulted in the most robust growth of tomato seedlings, indicating that GSA was particularly effective in alleviating the adverse effects of salt stress (Figure 1).

The salt stress significantly inhibited plant growth. The grafting and the exogenous SA promoted tomato growth under varying degrees of salt stress. Compared with the S treatment, the stem lengths increased by 22.5%, 26.5%, and 40.2% in the SA, G, and GSA treatments, respectively; the stem diameters increased by 11.0%, 26.0%, and 23.7%, respectively; the shoot fresh weights increased by 76.3%, 113.2%, and 247.4%, respectively; the root fresh weights increased by 150.9%, 238.6%, and 286.0%, respectively; the shoot dry weights increased by 53.5%, 65.1%, and 162.8%, respectively; the root dry weights increased by 150.0%, 150.0%, and 166.7%, respectively; and the leaf number increased by 0%, 10.0%, and 16.7%, respectively (Table 1). Among the four treatments, GSA significantly enhanced the stem length, stem diameter, and the shoot fresh weight compared with the other treatments.

### 2.2. Light Response Curve, Photosynthetic Parameters, and Chlorophyll Content

Compared with CK, the net photosynthetic rate (Pn) was markedly reduced across all other treatments as seen from the changes in the light response curve, indicating that the salt stress inhibited photosynthesis in plant leaves. However, under salt stress, both G and SA improved photosynthesis, with GSA exhibiting the most pronounced improvement (Figure 2A). Interestingly, the contents of chlorophyll a and chlorophyll b significantly increased under salt stress (0.86 and 0.49 mg/g, respectively), which was consistent across all the treatments. Nonetheless, the G and GSA treatments led to a decrease in the chlorophyll a and chlorophyll b levels, restoring them to levels comparable to those of CK (Figure 2B,C).

Table 2 illustrates that under salt stress, the photosynthetic rate of plants decreased significantly from 26.5% to 44.8% compared to that of CK, consistent with the results of the light response curve. The S treatment presented the lowest values of Pn, stomatal conductance (Gs), intercellular CO_2_ concentration (Ci), and transpiration rate (Tr). In contrast, the SA, G, and GSA treatments significantly improved these photosynthetic indices, with the most substantial enhancement observed in the GSA treatment.

### 2.3. Osmotic Substance

The soluble sugar content was the highest in the SA treatment, while it did not differ significantly from that of the CK and S treatments. However, in the grafted seedlings (G and GSA), the soluble sugar content decreased significantly (Figure 3A). The soluble protein content was the lowest in the S treatment, which was significantly different from other treatments and the highest in the GSA treatment (Figure 3B). The proline content exhibited the most pronounced changes among the treatments, with a significant increase under the salt stress. Notably, the GSA treatment demonstrated a significantly higher proline content than other treatments (Figure 3C). These findings highlighted the critical role of proline in regulating osmotic stress, surpassing the roles of soluble sugars and proteins.

### 2.4. Antioxidant Enzyme

The antioxidant enzyme activity was examined to further assess the effects of grafting and exogenous SA. As seen in the results presented in Figure 4, compared to CK, superoxide dismutase (SOD) activity increased while the catalase (CAT) activity decreased under S, and both SOD and CAT activity were significantly increased under SA, G, and GSA. There was no significant difference between SA and GSA, but both treatments significantly increased the SOD and CAT activity compared to G treatment.

### 2.5. Ionic Contents

The salt stress caused a significant increase in Na^+^ content, elevating it by 8–12 times in plants (Figure 5A), with a more pronounced increase in the self-rooted seedlings (S and SA) compared to grafted seedlings (G and GSA). In terms of K^+^ absorption, the SA, G, and GSA treatments had higher potassium ion contents than S, with G presenting significantly higher levels than other treatments (Figure 5B). The Ca^2+^ content was significantly higher in grafted seedlings (G and GSA) than in self-rooted seedlings (S and SA) and CK, with GSA exhibiting the highest Ca^2+^ content (Figure 5C). The trends in S^2−^ and Mg^2+^ content were similar, while the Mg^2+^ levels differed significantly between the G and GSA treatments (Figure 5D,E). The Al^3+^ content was the highest in the S treatment but did not significantly differ from SA, whereas the grafted seedling treatments (G and GSA) had significantly lower Al^3+^ content or levels than the CK (Figure 5F).

### 2.6. Transcriptomic Analysis

To thoroughly investigate the molecular mechanisms through which grafting and exogenous salicylic acid (SA) could aid tomato seedlings in coping with salt stress, we created five cDNA libraries, each consisting of three replicates: CK, the 150 mM NaCl-treated seedlings (S), the NaCl-treated grafted seedlings (G), the NaCl plus SA-treated seedlings (SA), and the NaCl plus SA-treated grafted seedlings (GSA). The transcriptome data analysis revealed 723,953,628 clean reads from the 15 libraries, with specific read counts of 146,306,824 for CK; 141,863,310 for S; 143,258,012 for G; 148,509,590 for SA; and 144,015,892 for GSA. All the libraries had a quality score (Q30) greater than 90%, indicating that the dataset was of high quality and suitable for in-depth analyses.

#### 2.6.1. Identification of Differentially Expressed Genes in Leaves among Treatments

We used DESeq2 [29] software (1.44.0) to normalize the gene counts in each sample, estimate the expression levels using the BaseMean values, calculate the fold changes, and perform the differential significance tests using a negative binomial distribution. Compared with CK, the S-treated tomatoes exhibited a total of 3674 differentially expressed genes, with 1860 genes up-regulated and 1814 genes down-regulated. The G-treated tomatoes had 4274 differentially expressed genes, with 2323 genes up-regulated and 1951 genes down-regulated. In the SA-treated tomatoes, there were 2606 differentially expressed genes, with 1103 genes up-regulated and 1503 genes down-regulated. The GSA-treated tomatoes had 3632 differentially expressed genes, with 1587 genes up-regulated and 2045 genes down-regulated. Compared to S, the SA treatment presented 1593 differentially expressed genes, with 656 genes up-regulated and 937 genes down-regulated. The GSA treatment resulted in 1027 differentially expressed genes, with 258 genes up-regulated and 769 genes down-regulated (Figure 6A).

The comparison of differentially expressed genes indicated that G vs. CK and S vs. CK had 1792 common differentially expressed genes, with 814 unique genes in the G treatment and 1882 unique genes in the S treatment. SA vs. CK and S vs. CK had 2791 common differentially expressed genes, with 1483 unique to the SA treatment and 883 unique to the S treatment. GSA vs. CK and S vs. CK had 2511 common differentially expressed genes, with 1121 unique genes in the GSA treatment and 1163 unique genes in the S treatment. GSA vs. CK and SA vs. CK had 2501 common differentially expressed genes, with 1131 unique to the GSA treatment and 1773 unique to the SA treatment (Figure 6B).

Compared to the S treatment, the SA and G treatments had 346 common differentially expressed genes, GSA and G had 356 common differentially expressed genes, and GSA and SA had 319 common differentially expressed genes. The SA treatment had 1063 unique differentially expressed genes, the G treatment had 496 unique differentially expressed genes, and the GSA treatment had 581 unique differentially expressed genes (Figure 6C).

#### 2.6.2. Gene Ontology (GO) Function Analysis of Differentially Expressed Genes among Different Treatments

The functions of DEGs can be categorized into biological processes (BP), cellular components (CC), and molecular functions (MF). The top 30 enriched pathways based on the number of up-regulated DEGs are detailed in Figure 7. For NaCl vs. CK, the most enriched GO terms in the BP category included flavonoid biosynthetic process, response to water deprivation, and response to abscisic acid; in the CC category, the terms included CCAAT-binding factor complex and integral component of membrane; and in the MF category, DNA-binding transcription factor activity was prominent. For G vs. CK, the top BP terms were flavonoid biosynthetic process, plant-type secondary cell wall biogenesis, and response to heat; the CC terms included integral component of the plasma membrane and CCAAT-binding factor complex; and MF terms were DNA-binding transcription factor activity, sequence-specific DNA binding, and heme binding. For SA vs. CK, the leading BP terms were flavonoid biosynthetic process, activation of immune response, and lipid transport; the CC terms were extracellular region, cell wall, and chromosomal region; and the MF terms included sequence-specific DNA binding, DNA-binding transcription factor activity, and heme binding. For GSA vs. CK, the top BP terms were response to water deprivation, flavonoid biosynthetic process, and response to abscisic acid; the CC terms included CCAAT-binding factor complex, mitochondrial alpha-ketoglutarate dehydrogenase complex, and chromosomal region; and the MF terms were DNA-binding transcription factor activity, sequence-specific DNA binding, and iron ion binding. For G vs. S, the leading BP terms were auxin-activated signaling pathway, regulation of growth, and response to biotic stimulus; the CC terms included extracellular region, nucleotide-excision repair factor 4 complex, and cell wall; and the MF terms were chitinase activity, Delta12-fatty-acid desaturase activity, and heme binding. For SA vs. S, the top BP terms were photosynthesis, light harvesting in photosystem I, protein–chromophore linkage, and response to light stimulus; the CC terms included photosystem I, photosystem II, and plastoglobuli; the MF terms were serine-type endopeptidase activity, chlorophyll binding, and 2 iron–2 sulfur cluster binding. For GSA vs. S, the top BP terms were protein folding and response to biotic stimulus; the CC terms included endoplasmic reticulum lumen; the MF terms were misfolded protein binding and unfolded protein binding.

#### 2.6.3. Kyoto Encyclopedia of Genes and Genomes (KEGG) Analysis of DEGs among Different Treatments

The top 20 DEGs annotated using the KEGG pathway database for various treatments are shown in Figure 8. For S vs. CK, the up-regulated DEGs were primarily associated with the plant hormone signal transduction and flavonoid biosynthesis. In G vs. CK, significant enrichment was observed in the plant hormone signal transduction and flavonoid biosynthesis. For SA vs. CK, notable enhancement was observed in the metabolic pathways, with significant enrichment in the plant hormone signal transduction and flavonoid biosynthesis. For GSA vs. CK, significant enrichment was noted in the plant hormone signal transduction and the protein processing in the endoplasmic reticulum. In the comparison of G vs. S, significant enrichment was observed in the plant hormone signal transduction and protein processing in the endoplasmic reticulum. For SA vs. S, considerable enhancement was observed in the photosynthesis-antenna proteins, photosynthesis, and plant hormone signal transduction. For GSA vs. S, considerable enhancement was observed in protein processing in the endoplasmic reticulum, glutathione metabolism, and phenylpropanoid biosynthesis.

### 2.7. Hormones

As shown in Figure 9, the S treatment significantly reduced the levels of salicylic acid (SA), jasmonic acid (JA), and indole-3-acetic acid (IAA), while markedly increasing the content of abscisic acid (ABA). Compared to S, the SA treatment notably elevated the endogenous SA levels. However, in the GSA treatment, the endogenous SA content was lower than that in the G treatment. The G treatment significantly increased the SA content and decreased the ABA content. Although the GSA treatment also increased endogenous SA content, it did not surpass the levels observed in the G treatment and resulted in a significant decrease in both the SA and JA content. Compared to S, GSA promoted IAA content significantly.

### 2.8. Confirmation of Differentially Expressed Genes by qRT-PCR Analysis

To validate the reliability of the transcriptomic data, we assessed the expression levels of ten genes associated with the plant hormone signaling transduction, secondary metabolite biosynthesis, and photosynthesis using qPCR. The qPCR results for these genes across the four treatments were largely aligned with the RNA-seq data, confirming the reliability of our transcriptome data (Figure 10).

### 2.9. Correlation Analysis

The correlation analysis shows a significant positive correlation between plant growth indicators. Additionally, growth indicators are significantly positively correlated with soluble protein, Pn, IAA, K^+^, and Ca^2+^. Conversely, they are significantly negatively correlated with ABA, Na^+^, and Al^3+^ (Figure 11). Pn is significantly positively correlated with the contents of IAA, JA, K^+^, S^2−^, and Mg^2+^ (Figure 11). To better understand the effects of various treatments on the physiological ecology of plants, principal component analysis was performed.

Principal component analysis (PCA) revealed the distribution and variation trends of various physiological parameters under different treatments (CK, S, G, SA, GSA). PCA was performed on 23 indicators, which can be mainly divided into four principal components (PCs). These components collectively explain 93.758% of the variance. Specifically, PC1 contributes 56.526% of the variance, PC2 accounts for 25.11%, PC3 explains 7.631%, and PC4 contributes 4.491% of the variance, collectively describing the changes in the physiological state of plants under different treatment conditions. In PC1, the variables with larger loadings are biomass indicators of the plants, ion balance, as well as Pn, ABA, and IAA (Figure 12). In PC2, physiological stress indicators such as soluble sugars, proteins, SOD, CAT, and hormones such as SA and JA occupy larger loadings (Figure 12). It can be seen that salt stress leads to an increase in Na^+^ accumulation and inhibits the absorption of K^+^ while promoting the accumulation of ABA content, thereby further inhibiting plant growth.

## 3. Discussion

### 3.1. Effect of Graft, Exogenous SA, and Their Interaction on Ionic Content of Tomato under Salt Stress

Ion stress is a critical factor in the salinity-induced stress in plants. Excess Na^+^ entering plant cells during salt stress can lower the membrane potential below its resting state, thereby activating the outward-rectifying potassium channels, such as guard cell outward-rectifying K^+^ channel, which results in potassium efflux [30,31]. Therefore, maintaining a balanced Na^+^/K^+^ ratio in the cell membrane is essential for salt tolerance [32]. Our experimental results revealed the significant increase and decrease in the sodium ion content and the potassium ion content (Figure 5) in plants under the salt stress, respectively, which was consistent with previous findings [33,34]. In contrast to the S treatment, the SA, G, and GSA treatments significantly reduced the Na^+^ content and increased the K^+^ content (Figure 5A,B). Notably, the G treatment exhibited the highest increase in the potassium content, indicating the potential of grafting to restore the ionic balance and enhance the salt tolerance. Previous research suggested that Na^+^ tends to accumulate near the root systems of self-grafted plants [35], whereas grafting can prevent excessive Na^+^ accumulation in the scion, thus reducing the ion toxicity in grafted seedlings [36]. Additionally, Ca^2+^ is an essential signaling element that has been shown to be involved in salt stress regulation [37]. The previous studies found that Ca^2+^ signaling triggered by salt stress first appeared at the root tip and then gradually spread to the root base [38,39]. Recently, it has been found that there is also an increase in Ca^2+^ content in leaves under salt stress. Our study demonstrated that in the grafting seedling treatments (G and GSA), the Ca^2+^ content was significantly higher than in other treatments (Figure 5C), likely due to the enhanced Ca^2+^ adsorption by the root system of the rootstock, which was consistent with previous findings [36,40,41]. The increase in calcium ion content helps resist salt stress, primarily through mechanisms involving the SOS pathway [42,43], ROS regulation [44], and ion uptake and homeostasis [45,46].

### 3.2. Effect of Graft, Exogenous SA, and Their Interaction on Osmotic Substances of Tomato under Salt Stress

The effective management of the electrical conductivity (EC) of nutrient solutions is essential in modern greenhouse cultivation because high EC levels can induce salt stress, typically due to the selective absorption at plant roots [47,48]. Plants exhibit various physiological responses to maintain an osmotic balance between internal and external cellular environments [49]. Under salt stress conditions, ion stress often occurs because of a substantial influx of Na^+^ into plant cells [50]. This increase in sodium ion concentration prompts plants to regulate their physiological activities and metabolism by adjusting the levels of osmotic regulators, such as soluble sugars, soluble proteins, and proline [51]. Our study discovered that under salt stress, the levels of soluble proteins and proline increased significantly in plants subjected to the SA, G, and GSA treatments (Figure 3B,C), consistent with previous findings [6,52,53]. The change in the proline content was notably more pronounced than that in the soluble proteins and sugars, indicating that plants primarily utilized the proline as the key substance for stress resistance against environmental pressures. The sodium-induced salt stress resulted in an increase in the proline content and a reduction in plant water potential [54,55]. Furthermore, under the GSA treatment, the proline content was significantly higher than that in other treatments, suggesting a synergistic effect of grafting and exogenous SA in promoting proline accumulation. This enhanced proline accumulation may be attributed to the increased activity of proline synthetase induced by exogenous SA [52].

### 3.3. Effect of Graft, Exogenous SA, and Their Interaction on Antioxidant Enzyme of Tomato under Salt Stress

Earlier studies have shown that under salt stresses, they activate salt tolerance mechanisms that enhance antioxidant enzyme activity to remove harmful ROS and free radicals [56]. SOD primarily removes the accumulation of reactive oxygen species caused by salt stress, while CAT decomposes hydrogen peroxide, thereby protecting the cells [57]. In this study, SOD significantly increased under salt stress (Figure 4A), which is consistent with previous research [58], while CAT levels significantly decreased under S treatment (Figure 4B), which is believed to be related to the serious growth inhibition in the salt stress treatment. Furthermore, SOD and CAT activity were significantly increased under G, SA, and GSA treatments, which is consistent with previous studies [6,36]. Among these, the enzyme activity was highest under the GSA treatment (Figure 4), likely due to the synergistic effect of grafting and exogenous SA combined treatment, resulting in an additive effect.

### 3.4. Effect of Graft, Exogenous SA, and Their Interaction on Hormones of Tomato under Salt Stress

Phytohormones are small chemical compounds crucial for the regulation of plant growth and development [59]. They coordinate complex processes across different plant stages and tissues. ABA, SA, and JA are commonly classified as stress-response hormones because of their roles in mediating plant responses to stress [60]. In this experiment, various treatments led to significant changes in the endogenous SA content. The lowest levels of the endogenous SA were observed under the S treatment, whereas treatments involving the exogenous SA, grafting, or the combination of both increased the endogenous SA content, with the highest increase observed in the G treatment (Figure 9A). This increase in the endogenous SA could enhance the plant’s defense mechanisms against salt stress by strengthening the antioxidant system and promoting photosynthesis [61]. However, compared with the G treatment, the endogenous SA levels were significantly reduced in the GSA treatment group (Figure 9A). This decrease may be due to a dose-dependent response, where the exogenous SA application affected the endogenous SA synthesis, potentially weakening the salt stress mitigation effect when the SA concentrations were either too high or too low [62,63,64].

The ABA content increased rapidly in plants under salt stress to mitigate injury by regulating the stomatal opening, water evaporation, and other physiological activities, thus serving as an indicator of the degree of salt stress injury. The results of this experiment showed that ABA content significantly decreased in G, SA, and GSA treatments, compared to salt stress alone (Figure 9B), suggesting that the plants under the G, SA, and GSA treatments experienced less damage from salt stress. This is because grafting enhanced the ability of the root system to absorb water and nutrients [65]. Additionally, exogenous SA can regulate the ion balance and remove the accumulation of reactive oxygen species [66].

### 3.5. Effect of Graft, Exogenous SA, and Their Interaction on Photosynthesis of Tomato under Salt Stress

Salt stress can adversely affect plant growth and productivity by disrupting key physiological processes, particularly photosynthesis. The accumulation of sodium ions in plant tissues under salt stress can disrupt the sodium–potassium ratio, which is a critical factor influencing the changes in photosynthesis [67]; this can be consistent with the results of this study. Plants have inherent stress resistance and typically employ physiological adaptations to mitigate adverse effects. Previous studies demonstrated that plants close their stomata in response to drought stress to conserve water, thereby increasing the water use efficiency (WUE) [14,68]. Our study indicated that the plants under salt stress exhibited similar stress responses, leading to reduced photosynthesis (Figure 2 and Table 2). SA is known to regulate the potassium channels and promote the stomatal opening [69], aligning with the results of this study. In addition, ABA regulates the stomatal behavior by inducing the stomatal closure to reduce water loss. However, in this study, the ABA content was significantly reduced across all three treatments (SA, G, and GSA), which promoted the stomatal opening and potentially enhanced photosynthesis. This reduction in ABA levels could contribute to the improved photosynthesis observed under these treatments, providing a physiological mechanism through which the plants could respond to salt stress by modulating the stomatal behavior.

### 3.6. Effect of Graft, Exogenous SA, and Their Interaction on Transcriptome of Tomato under Salt Stress

Based on those results, grafting combined with exogenous SA could regulate the plant hormone levels, maintain the cellular osmotic balance by modulating the proline and ionic contents, and promote photosynthesis and root development in tomato seedlings. To further investigate the effects of grafting and SA on tomato growth under salt stress, we conducted the transcriptomic analysis of the tomato leaves under various treatments. The transcriptome analysis revealed that grafting, exogenous SA, or their combination (GSA) regulated tomato growth under salt stress by affecting the phytohormone signaling pathway, the MAPK signaling pathway, and glutathione metabolism. Compared to the S treatment, the exogenous SA increased the expression of the endogenous SA synthesis gene (*solyc01g106605.1*), and the grafting up-regulated the *solyc09g007010.1* gene, both contributing to a significant increase in the endogenous SA content (Figure 9A). Additionally, the exogenous SA down-regulated the expression of *PP2CA*, *PP2C51*, and *ABI5*, which were related to the ABA synthesis [70], leading to the decreased ABA content (Figure 9B). These findings suggested that the exogenous SA and grafting enhanced the plant growth under salt stress by modulating the phytohormone-related gene expression. Furthermore, the exogenous SA up-regulated the genes associated with the chloroplast and antenna proteins (*LHCA*), enhancing the efficiency of photosystems I and II, improving electron transport, and increasing the plant’s light-harvesting capacity. This response resulted in improved photosynthesis in tomatoes under salt stress (Figure 2, Table 2). In this study, under the GSA treatment, the plant hormone signal transduction pathway was significantly enriched, *TIR1* and *IAA* genes were up-regulated, promoted IAA synthesis, and increased its content in leaves (Figure 9D), and further analysis revealed that genes such as *AHP/TF* and *SnRK2*, were also significantly up-regulated. These results indicated that hormonal regulation is a primary regulatory pathway for plants to resist salt stress.

## 4. Materials and Methods

### 4.1. Plants Materials and Growth Condition

The experiment was conducted in a modern greenhouse located in Chongming District, Shanghai, China, by Youyou Agricultural Technology Co., Ltd.

The tomato cultivars Jiaxina and Maxifort were selected as the scions and rootstocks, respectively. Jiaxina (Rijk Zwaan, De Lier, The Netherlands) is a tomato cultivar in greenhouses, known for its good commercial fruit quality, especially for the Tomato Yellow Leaf Curl Virus (TYLCV) resistance. Maxifort (Bayer, Leverkusen, Germany) is a popular and widely used commercial rootstock variety, with a well-developed root system and good grafting compatibility. The seeds of Jiaxina and Maxifort were sowed in rockwool plugs; then we made sure that plugs in the trays (240 plugholes) were well watered with EC-level 1.0–1.5 mS cm^−1^ and pH 5.5–6.0 of nutrient solution. Grafting was performed when the seeding stem diameter was approximately 2 mm (15 d after sowing). The Jiaxina cultivar was divided into two groups: one was grafted onto Maxifort rootstock, while the other underwent self-grafting (Jiaxina × Jiaxina). The rootstock seedlings were cut about 1 cm below the cotyledon leaves, and scion seedlings were cut about 0.5 cm above the cotyledonary leaves. Grafts were made immediately after cutting the plants, and grafting clips were used to adhere to the graft union. The grafted tomato seedlings were cultivated in a plant factory with artificial light at a temperature of approximately 25 °C and 100% relative humidity. After the initial recovery period of 4 d, the plants were moved to a cultivation chamber [19]. Following another seven days, the seedlings were transplanted into the coconut fiber blocks (Van der Knaap, Kwintsheul, the Netherlands). The transplanted seedlings were then placed in the glasshouse (day temperature about 24–28 °C, night temperature about 16–18 °C, and ambient light condition) for one week to acclimate and grow before the commencement of the experimental procedures.

### 4.2. Application of Sodium Chloride (NaCl) and Salicylic Acid (SA)

Salt stress treatment began on 18 January 2023 (18 d after grafting). The self-rooted tomato seedlings and those grafted onto rootstocks were exposed to 150 mM NaCl to establish the S (the self-rooted seedlings irrigated the nutrient solution (pH = 6.0, EC = 2.0 mS cm^−1^) with 150 mM NaCl) and G (the grafted seedlings irrigated nutrient solution (pH = 6.0, EC = 2.0 mS cm^−1^) with 150 mM NaCl) treatments, respectively. To investigate the effects of exogenous salicylic acid (SA) on plant stress response, an additional treatment group was introduced, where 0.2 mM SA was administered via root irrigation, denoted as SA (the self-rooted seedlings irrigated nutrient solution (pH = 6.0, EC = 2.0 mS cm^−1^) with 150 mM NaCl and 0.2 mM SA.) and GSA (the grafted seedlings irrigated nutrient solution (pH = 6.0, EC = 2.0 mS cm^−1^) with 150 mM NaCl and 0.2 mM SA) treatments. The CK treatment consisted of the self-rooted seedlings irrigated with a nutrient solution (pH = 6.0, EC = 2.0 mS cm^−1^). The treatments were applied every alternate day. The schematic representation of the treatment groups is shown in Figure 13. Each treatment was replicated three times, with each replicate consisting of six individual plants to ensure statistical reliability.

### 4.3. Salinity Maintenance and Irrigation

The entire duration of the salt stress treatment was 10 days. Throughout the experiment, the tomato plants were transplanted into the coconut fiber blocks and grown in cavity trays, with their root systems irrigated daily with various solutions. These solutions included a NaCl solution and a NaCl solution containing exogenous salicylic acid. The roots were soaked in these solutions for one hour to allow the plants to absorb the desired nutrients and components, after which any excess solution was drained off. This procedure ensured consistent exposure to the treatments while preventing issues related to overwatering or stagnant moisture.

### 4.4. Determination Methods

#### 4.4.1. Morphological Parameters

Morphological parameters were measured on the 10th day. The stem length was measured from the base of the shoot to the growing point using a ruler, and the average was calculated in centimeters. The stem diameters were measured 2 cm above the incision with a digital Vernier caliper, and the average was calculated in mm. The number of leaves was recorded by counting the total number of leaves, and the average was computed. The root length, leaf length, and leaf diameter were measured using a ruler, and averages were calculated in cm. The measurements were repeated at least three times. Five plants from each treatment group were randomly selected and harvested destructively. The fresh weights of the shoots and roots were determined, followed by drying for at least 48 h at 105°C in a ventilated oven to measure the dry weights.

#### 4.4.2. Leaf Gas Exchange Parameters

The photosynthetic parameters were assessed on 28 January using a CIRAS-3 portable photosynthesis system (PP Systems, Amesbury, MA, USA). Before the measurements, the plants were allowed a minimum of 30 min to adapt to the measurement conditions. The leaves were placed in a 4.5 cm^2^ conditioning chamber, where a light source provided a consistent photon flux density of 1000 μmol m^−2^ s^−1^. Various parameters, including Pn, Ci (intercellular CO_2_ concentration), Gs (stomatal conductance), Tr (transpiration rate), and WUE, were determined. The photosynthetic light response curve was established by gradually increasing the photon flux density from approximately 0 to 10, 20, 50, 80, 100, 120, 150, 200, 400, 600, 800, 1000, 1200, 1500, and 1800 μmol m^−2^ s^−1^, respectively. The air temperature, the CO_2_ concentration, and the relative humidity were adjusted to match the greenhouse conditions. The measurements were repeated three times to ensure reliability and consistency.

#### 4.4.3. Osmotic Substance

On the 10th day of the salt stress treatment, destructive sampling was performed. The soluble sugar and soluble protein content were measured using a previously described method [71]. The proline content was determined using assay kits (Shanghai Yuanye Bio-Technology Co., Ltd., Shanghai, China) following the manufacturer’s instructions. The measurements were repeated three times for each treatment.

#### 4.4.4. Antioxidant Enzyme

Samples were taken on the 10th day of treatment. The SOD and CAT activities were measured using test kits purchased from Suzhou Omin Biotechnology Co., Ltd. (Suzhou, China). The measurements were repeated three times for each treatment.

#### 4.4.5. Ion Content

The plant samples were collected on the 10th day. Then the samples were dried in a ventilated oven for 48 h and then ground into a powder using a grinder (IKA Works GmbH & Co., Staufen, Germany). Samples (10 g) were weighed to measure the mineral element content. The measurement results were provided by Shanghai OE Biotech Co., Ltd. (Shanghai, China). The measurements were repeated three times for each treatment.

#### 4.4.6. Phytohormone

Phytohormone sampling was conducted on the 10th day. The phytohormones were detected using ultra-performance liquid chromatography–electrospray ionization tandem mass spectrometry (UPLC–ESI–MS/MS) at the Shanghai Luming Biotechnology Co., Ltd. (Shanghai, China). The samples were prepared by grinding 50 mg of each treatment group into a powder. The mixture of 2-propanol/H_2_O/concentrated HCl (2:1:0.002, *v*/*v*) was used as the extraction solvent. The sample solution was analyzed using the reverse-phase C18 Gemini HPLC column coupled with HPLC–ESI–MS/MS (AB SCIEX, Framingham, MA, USA). The experimental conditions and setup for HPLC–ESI–MS/MS and multiple reaction monitoring (MRM) followed previously described protocols [72]. The measurements were repeated three times for each treatment.

#### 4.4.7. Transcriptome Analysis

After the 10-day salt stress treatment, the plant samples were destructively harvested. Accurate 0.5 g samples were weighed and immediately frozen with liquid nitrogen. Three biological replicates were used for each treatment. The total RNA was extracted from the treatment groups using the TRIzol reagent (Invitrogen, Carlsbad, CA, USA). The RNA purity and quantification were assessed with a NanoDrop 2000 spectrophotometer (Thermo Scientific, Waltham, MA, USA), and the RNA integrity was evaluated using an Agilent 2100 Bioanalyzer (Agilent Technologies, Santa Clara, CA, USA). The libraries were constructed following the VAHTS Universal V6 RNA-seq Library Prep Kit instructions, and the transcriptome sequencing and analysis were performed by OE Biotech Co., Ltd. (Shanghai, China).

### 4.5. Data Analysis

To ensure data accuracy, each treatment was assessed at least three times. The mean values and standard errors were calculated using Microsoft Excel Mondo 2016 (Microsoft, Washington, DC, USA). The significance of mean differences (*p* < 0.05) was evaluated using Duncan’s test with SPSS software version 22.0 (IBM Corporation, New York, NY, USA). The figures were generated using Origin 2022 (OriginLab Corporation, Massachusetts, MA, USA).

## 5. Conclusions

The grafting and exogenous salicylic acid (SA) interacted to up-regulate genes associated with chloroplast and antenna proteins, enhancing the stomatal conductance (gs) and significantly increasing the net photosynthetic rate (Pn). This combination also promoted proline accumulation and maintained cellular osmotic stability; it also had significant enhancing effects on antioxidant enzyme activity. Additionally, the grafting and exogenous SA facilitated the potassium (K^+^) and calcium (Ca^2+^) uptake while reducing sodium (Na^+^) and aluminum (Al^3+^) accumulation, thereby mitigating ionic toxicity. The ionic regulation was crucial for preserving the plant health under salt stress conditions. Furthermore, these treatments modulated the hormone signaling pathways by up-regulating the endogenous SA synthesis genes and down-regulating the ABA synthesis-related genes, which increased the endogenous SA content and decreased the ABA content, thereby aiding the plant’s salt stress response (Figure 14). In conclusion, both exogenous SA and grafting effectively alleviated the salt stress, and the combined application further enhanced the stress mitigation through multiple mechanisms, including ion balance and hormone regulation, increased antioxidant enzyme activity, and osmotic adjustment. This synergistic approach could be promising for improving the resilience and productivity of plants in saline environments.

## Figures and Tables

**Figure 1 ijms-25-10799-f001:**
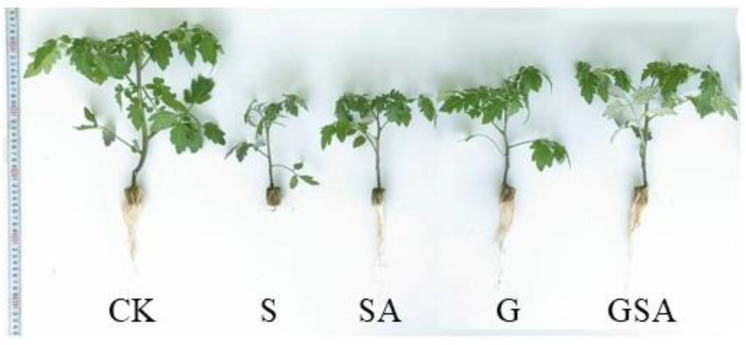
Differences in the growth changes of tomatoes under different treatments. CK: tomato seedling growth under normal conditions. S: NaCl-treated seedlings; G: NaCl-treated grafted seedlings; SA: NaCl plus SA-treated seedlings; GSA: NaCl combined with SA-treated grafted seedlings.

**Figure 2 ijms-25-10799-f002:**
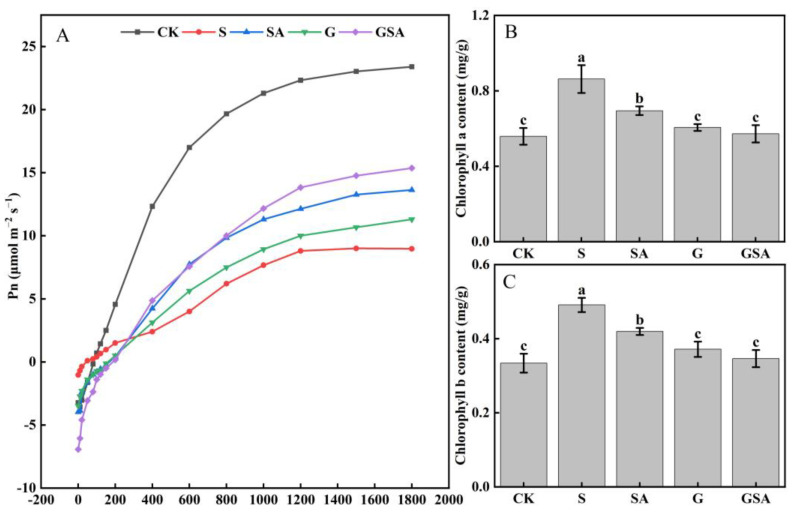
Light response curve (**A**), chlorophyll a content (**B**), and chlorophyll b content (**C**) of tomato leaves as affected by grafting and SA treatments under salt stress. Different letters indicate significant differences in the same parameter between treatments at *p* < 0.05.

**Figure 3 ijms-25-10799-f003:**
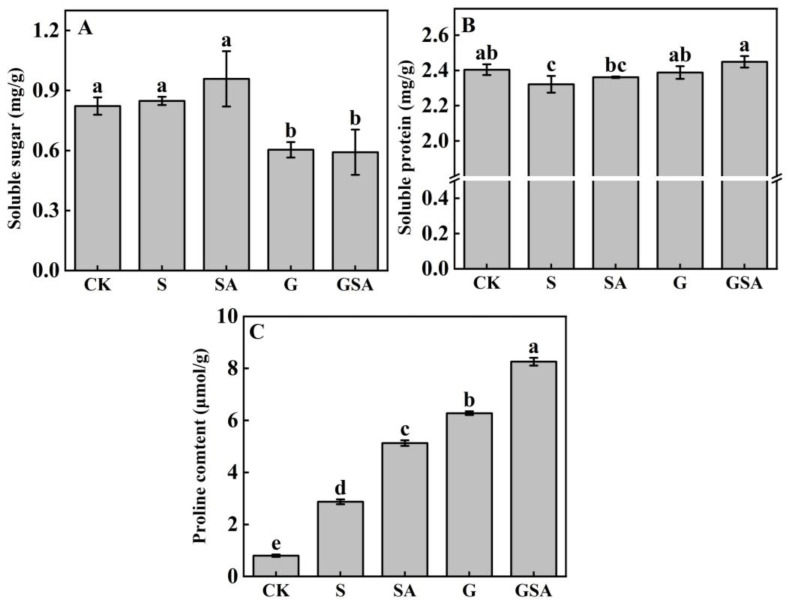
Soluble sugar (**A**), soluble protein (**B**), and proline (**C**) contents of tomato leaves as affected by salt stress (S), grafting (G), exogenous SA (SA), and grafting combined with exogenous SA (GSA) treatments under salt stress. CK: control treatment. Different letters indicate significant differences in the same parameter between treatments at *p* < 0.05.

**Figure 4 ijms-25-10799-f004:**
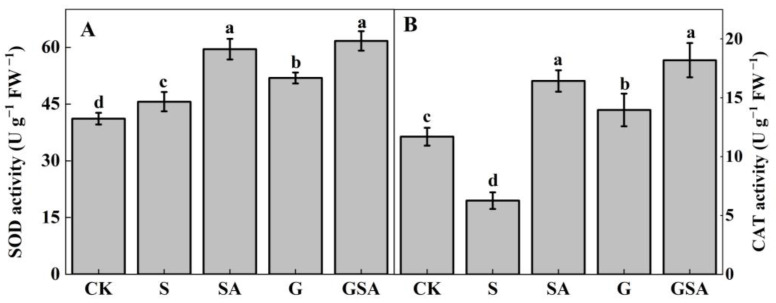
SOD (**A**) and CAT (**B**) activity of tomato leaves as affected by salt stress (S), grafting (G), exogenous SA (SA), and grafting combined with exogenous SA (GSA) treatments under salt stress. CK: control treatment Different letters indicate significant differences in the same parameter between treatments at *p* < 0.05.

**Figure 5 ijms-25-10799-f005:**
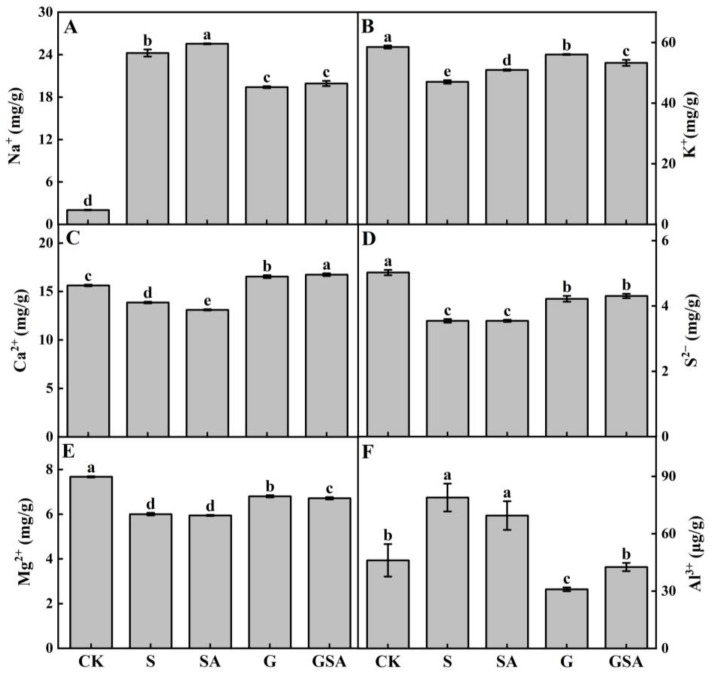
Na^+^ (**A**), K^+^ (**B**), Ca^2+^ (**C**), S^2−^ (**D**), Mg^2+^ (**E**), and Al^3+^ (**F**) contents of tomato as affected by salt stress (S), grafting (G), exogenous SA (SA), and grafting combined with exogenous SA (GSA) treatments under salt stress. CK: control treatment. Different letters indicate significant differences in the same parameter between treatments at *p* < 0.05.

**Figure 6 ijms-25-10799-f006:**
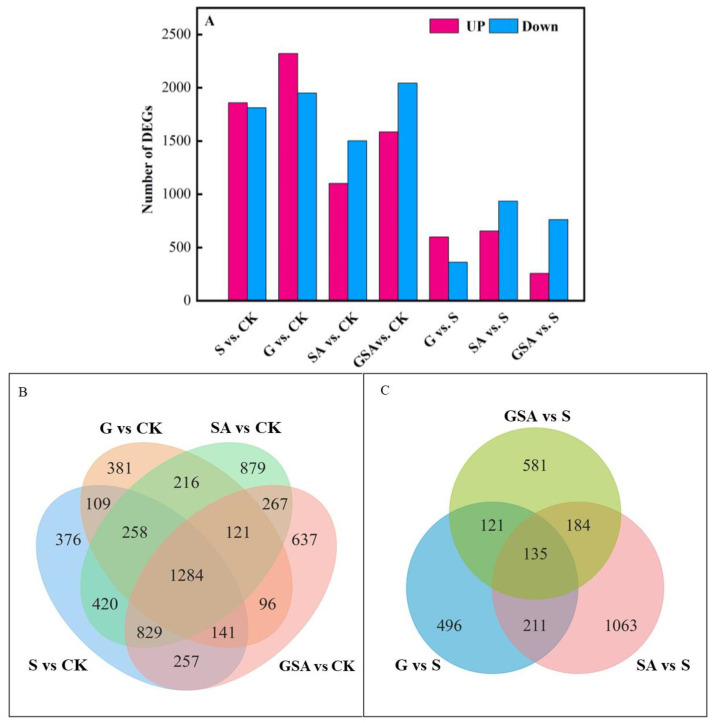
Transcriptomic data analysis of tomato seedlings under different treatments: (**A**) number of differentially expressed genes (DEGs) in S vs. CK, G vs. CK, SA vs. CK, GSA vs. CK, G vs. S, SA vs. S, and GSA vs. S comparisons; (**B**,**C**) Venn diagram of DEGs; CK, control treatment; S, NaCl treatment; SA, exogenous SA treatment; G, grafting treatment; GSA, combined grafting and exogenous SA treatment.

**Figure 7 ijms-25-10799-f007:**
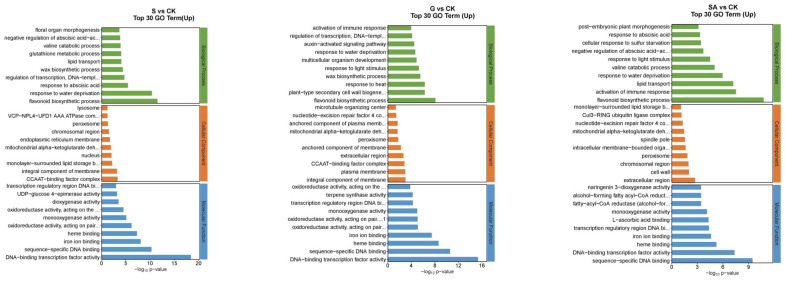
Top 30 GO annotations with the highest enrichment of DEGs in tomatoes under different treatments. CK, control treatment; S, NaCl treatment; SA, exogenous SA treatment; G, grafting treatment; GSA, combined grafting and exogenous SA treatment.

**Figure 8 ijms-25-10799-f008:**
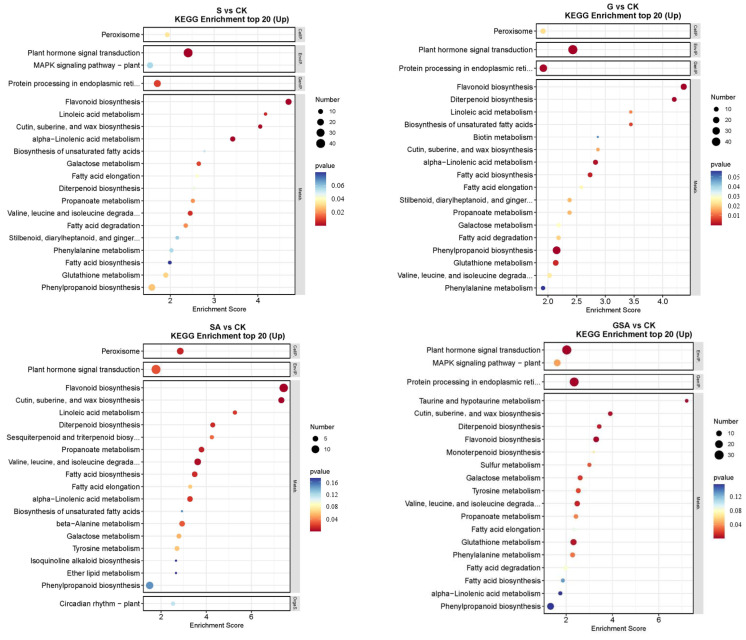
KEGG enrichment top 20 in tomatoes under different treatments. CK, control treatment; S, NaCl treatment; SA, exogenous SA treatment; G, grafting treatment; GSA, combined grafting and exogenous SA treatment.

**Figure 9 ijms-25-10799-f009:**
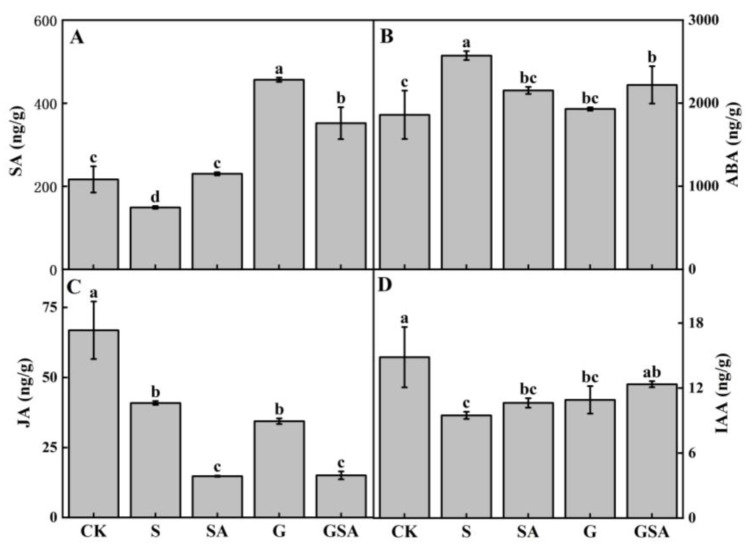
Endogenous SA (**A**), ABA (**B**), JA (**C**), and IAA (**D**) contents of tomato as affected by salt stress (S), grafting (G), exogenous SA (SA), and grafting combined with exogenous SA (GSA) treatments under salt stress. CK: tomato seedling growth under normal conditions. Different letters indicate significant differences in the same parameter between treatments at *p* < 0.05.

**Figure 10 ijms-25-10799-f010:**
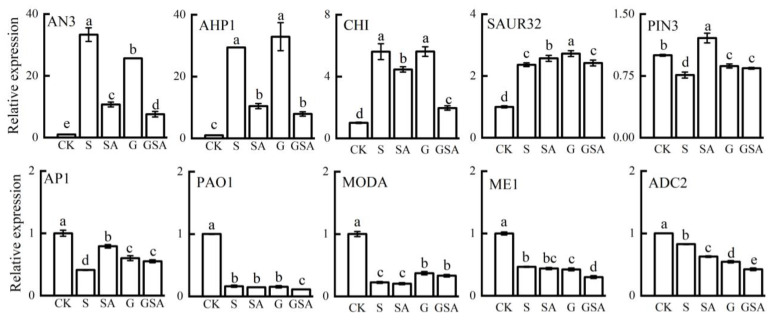
A comparison of the expression patterns of 10 candidate genes between RNA-seq and qRT-PCR for selected transcripts. Data represent the mean ± SD of three independent experiments. Different lowercase letters indicate significant differences among treatments at *p* < 0.05.

**Figure 11 ijms-25-10799-f011:**
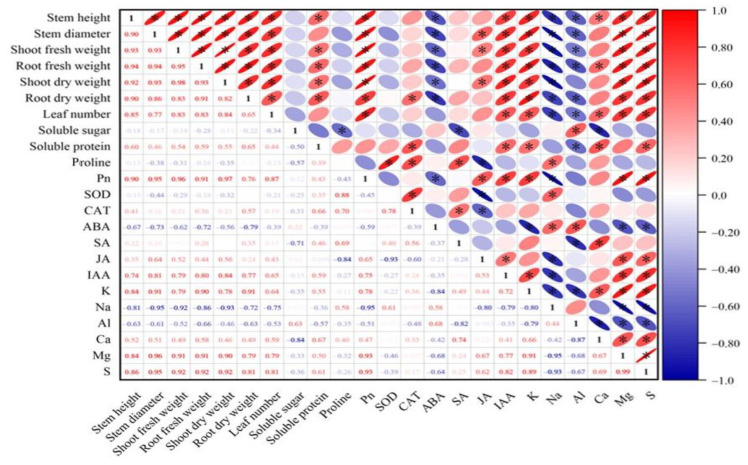
Correlation analysis between different results. Asterisks indicate that there is a correlation between the results. Asterisk (*) represents significant correlation between two results, red represents positive correlation, blue represents negative correlation.

**Figure 12 ijms-25-10799-f012:**
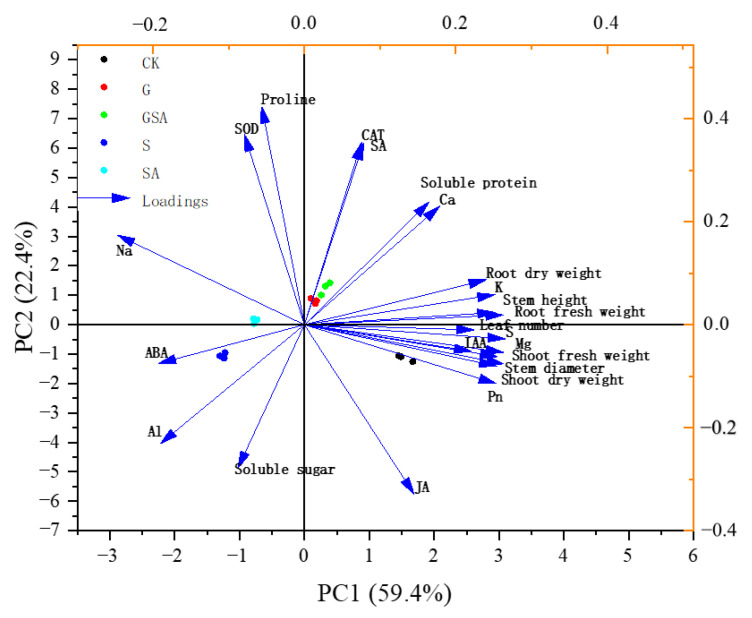
Principal component analysis of osmotic adjustment substance and enzyme activity, ion distribution-related parameters, hormone content, and plant growth characteristics. Arrow direction and length indicate correlation and strength, respectively. CK, control treatment; S, NaCl treatment; SA, exogenous SA treatment; G, grafting treatment; GSA, combined grafting and exogenous SA treatment.

**Figure 13 ijms-25-10799-f013:**
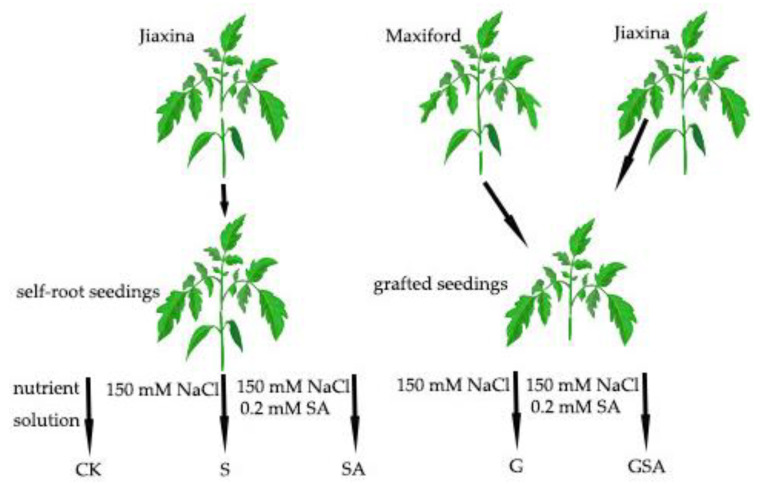
A schematic diagram of the experimental process. Note: NaCl and SA were prepared using the nutrient solution (pH = 6.0, EC = 2.0 mS cm^−1^) as solvent. CK, the self-rooted seedlings irrigated the normal nutrient solution. S, the self-rooted seedlings irrigated nutrient solution with 150 mM NaCl. SA, the self-rooted seedlings irrigated nutrient solution with 150 mM NaCl and 0.2 mM SA. G, the grafted seedlings irrigated nutrient solution with 150 mM NaCl. GSA, the grafted seedlings irrigated nutrient solution with 150 mM NaCl and 0.2 mM SA.

**Figure 14 ijms-25-10799-f014:**
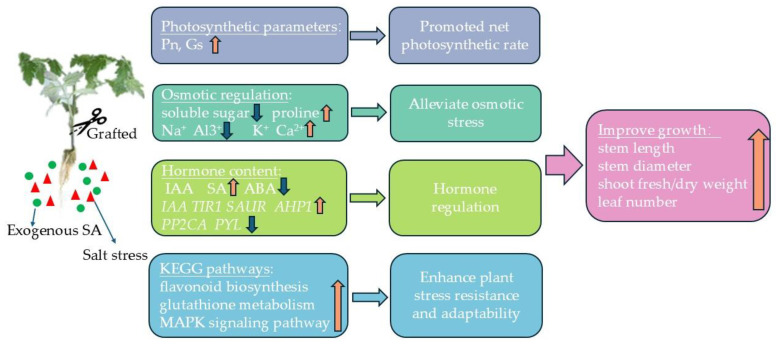
GSA regulates the physiological and molecular patterns of tomato seedlings.

**Table 1 ijms-25-10799-t001:** Effects of graft, exogenous SA, and their interaction on tomato morphology under salt stress.

Treatment	Stem Length/cm	Stem Diameter/mm	Shoot Fresh Weight/g	Root Fresh Weight/g	Shoot Dry Weight/g	Root Dry Weight/g	Leaf Number
CK	16.3 ± 0.8 a	6.86 ± 0.11 a	21.3 ± 2.2 a	3.27 ± 0.30 a	1.80 ± 0.11 a	0.21 ± 0.022 a	7.6 ± 0.5 a
S	10.2 ± 0.8 d	4.27 ± 0.24 d	3.8 ± 0.4 d	0.57 ± 0.12 d	0.43 ± 0.03 d	0.06 ± 0.002 c	6.0 ± 0.0 c
SA	12.5 ± 0.5 c	4.74 ± 0.06 c	6.7 ± 0.5 c	1.43 ± 0.21 c	0.66 ± 0.04 c	0.15 ± 0.009 b	6.0 ± 0.0 c
G	12.9 ± 0.8 c	5.38 ± 0.11 b	8.1 ± 0.9 c	1.93 ± 0.21 b	0.71 ± 0.08 c	0.15 ± 0.004 b	6.6 ± 0.5 b
GSA	14.3 ± 0.6 b	5.28 ± 0.13 b	13.2 ± 1.5 b	2.20 ± 0.26 b	1.13 ± 0.05 b	0.16 ± 0.005 b	7.0 ± 0.0 b

Different letters indicate significant differences (Duncan’s test, *p* < 0.05).

**Table 2 ijms-25-10799-t002:** Effects of graft and exogenous SA on photosynthetic parameters under salt stress.

Treatment	Pn (µmol CO_2_·m^−2^·s^−1^)	Gs (mmol H_2_O·m^−2^·s^−1^)	Ci (µmol CO_2_·mol^−1^)	Tr (mmol H_2_O·m^−2^·s^−1^)
CK	18.8 ± 1.5 a	254.3 ± 19.2 a	282.3 ± 8.1 a	3.21 ± 0.23 a
S	10.4 ± 0.7 d	61.0 ± 12.2 d	147.6 ± 37.3 d	1.27 ± 0.27 c
SA	10.8 ± 0.5 cd	100.3 ± 19.8 bc	243.0 ± 26.5 ab	1.94 ± 0.25 b
G	12.1 ± 0.7 c	80.6 ± 12.6 cd	190.0 ± 18.7 c	1.27 ± 0.18 c
GSA	13.8 ± 0.2 b	115.3 ± 4.9 b	225.0 ± b10.8 c	1.72 ± 0.03 b

Different lowercase letters indicate significant differences in the same parameter among treatments at *p* < 0.05.

## Data Availability

The data presented in this study are available upon request from the corresponding author.

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
