# Peer review of "An Enhanced Interaction of Graft and Exogenous SA on Photosynthesis, Phytohormone, and Transcriptome Analysis in Tomato under Salinity Stress"

_ijms, 2024, doi:10.3390/ijms251910799_

Round 1

Reviewer 1 Report

Comments and Suggestions for Authors

This study“An enhanced interaction of graft and exogenous SA on photo-synthesis, phytohormone and transcriptome analysis in tomato salinity stress” discusses the alleviating effects and mechanisms of grafting, exogenous salicylic acid (SA), and their combined use on tomato seedlings under salt stress. The topic is investigated in the literature, and there is a very few of reference published. However, this paper gives significant contribution to the current knowledge in related field. The data are sound and it deserves to be published, after minor revisions.

Overall Recommendations: Minor Revisions

Point 1: In Table 1, shoot dry weight and root dry weight are missing units.

Point 2: Please cite Figure No. or Table No. in brackets at suitable places for a better connectivity in results and discussion sections as to facilitate the reader.

Point 3: The text has many typing and grammatical errors, capitalization issues. English style and language require a profound revision. However, the readability of the manuscript needs to be improved.

Point 4: References formatting are inconsistent. It must be according to MDPI IJMS Journal. A few DOI missing.

Comments on the Quality of English Language

The text has many typing and grammatical errors, capitalization issues. English style and language require a profound revision. However, the readability of the manuscript needs to be improved.

Author Response

Point 1: In Table 1, shoot dry weight and root dry weight are missing units.

Thanks for pointing out the mistakes. This has been addressed in table 1.

Point 2: Please cite Figure No. or Table No. in brackets at suitable places for a better connectivity in results and discussion sections as to facilitate the reader.

Thank you for your guidance. Now we have cited Figure No. or Table No. in the discussion and results sections to enhance readability in lines 324, 336, 337,  360, 370, 386, 402, 403, 403, 417, 421, 429.

.

Point 3: The text has many typing and grammatical errors, capitalization issues. English style and language require a profound revision. However, the readability of the manuscript needs to be improved.

Thank you. We have asked the editing company Mjeditor (www.mjeditor.com) to revise the English and editing in the new version.

Point 4: References formatting are inconsistent. It must be according to MDPI IJMS Journal. A few DOI missing.

Thank you for your advice. We have carefully reviewed the reference format according to the journal's requirements and added the missing DOIs for some references in lines 718, 753, 792, 810.

Reviewer 2 Report

Comments and Suggestions for Authors

The present manuscript investigates the effect of exogenous application of salicylic acid on tomato plants growing under salinity conditions. Considering the fact that salinity is the second most important stressor on Earth. The use of natural substances as anti-stress agents is topical. The manuscript is relatively carefully written, but I still recommend some additions and modifications. I recommend adding a list of abbreviations used in the text, as not all of them are explained. The abstract is adequate, but it would still be useful to indicate, for example, by what percentage the characteristics mentioned have decreased or increased. The results are supported by graphs and tables. Unfortunately, some of the graphs are small and therefore do not have the necessary explanatory power, especially graphs 2, 3, 6 and 7. It is necessary to use those values that are in the graphs. The individual results are evaluated separately; it may be appropriate to add correlation analyses and correlations, e.g. PCA analysis. The discussion is in some parts rather just a statement of fact. Please check it. In the methodology, please explain the concept of what is a standard nutrient solution, see line 435? A variety of solutions are used. Has the nutrient solution been modified in any way after the salt has been added so as not to alter its properties, especially its water potential? What is the origin of plants? What was the light and temperature regime of the experiment? Was it natural light? This is the basis for setting parameters for measuring photosynthetic characteristics. Please clarify this. It is necessary to check the literature review used. It is not cited uniformly. 

Author Response

Point 1: I recommend adding a list of abbreviations used in the text, as not all of them are explained.

Thank you for your suggestion, the abbreviation list has already been added to the article in lines 649-680.

Point 2: The abstract is adequate, but it would still be useful to indicate, for example, by what percentage the characteristics mentioned have decreased or increased.

Thanks. This has already been resolved in lines 17-22.

Point 3: The results are supported by graphs and tables. Unfortunately, some of the graphs are small and therefore do not have the necessary explanatory power, especially graphs 2, 3, 6 and 7. It is necessary to use those values that are in the graphs.

Thank you for your suggestion. Figure 2 has been enlarged, and the data is now displayed directly in the text (line 131). Figure 3 has been redone and enlarged, with the data presented more clearly. Figures 6 and 7 have also been enlarged, and the key information from the figures has been included in the text.

Point 4: The individual results are evaluated separately; it may be appropriate to add correlation analyses and correlations, e.g. PCA analysis.

Thank you for your valuable suggestion. We have completed the PCA and correlation analyses in lines 320-347 “The correlation analysis shows that there is a significant positive correlation between plant growth indicators. Additionally, growth indicators are significantly positively correlated with soluble protein, Pn, IAA, K+ and Ca2+. Conversely, they are significantly negatively correlated with ABA, Na+, and Al3+ (Figure 11). Pn is significantly positively correlated with the contents of IAA, JA, and K+, S2-, Mg2+ (Figure 11). To better understand the effects of various treatments on the physiological ecology of plants, Principal Component Analysis was performed.

Principal Component Analysis (PCA) revealed the distribution and variation trends of various physiological parameters under different treatments (CK, S, G, SA, GSA). PCA was performed on 23 indicators, which can be mainly divided into 4 principal components (PCs). These components collectively explain 93.758% of the variance. Specifically, PC1 contributes 56.526% of the variance, PC2 accounts for 25.11%, PC3 explains 7.631%, and PC4 contributes 4.491% of the variance, collectively describing the changes in the physiological state of plants under different treatment conditions. In PC1, the variables with larger loadings are biomass indicators of the plants, ion balance, as well as Pn, ABA, and IAA (Figure 12). In PC2, physiological stress indicators such as soluble sugars, proteins, SOD, CAT, and hormones like SA and JA occupy larger loadings (Figure 12). It can be seen salt stress leads to an increase in Na+ accumulation and inhibits the absorption of K+, while promoting the accumulation of ABA content, thereby further inhibiting plant growth.

Point 5: The discussion is in some parts rather just a statement of fact. Please check it.

Thank you for your constructive feedback. We have reviewed the text and added explanations to better contextualize the factual statements in lines 430-433 “This is because grafting enhanced the ability of root system to absorb water and nutrients [65]. Additionally, exogenous SA can regulate the ion balance and remove the accumulation of reactive oxygen species [66].” Furthermore, we finished the discussion about SOD and CAT activity in lines 396-408 “Earlier studies have shown that under salt stresses, they activate salt tolerance mechanisms that enhance antioxidant enzyme activity to remove harmful ROS and free radicals [56]. SOD primarily removes the accumulation of reactive oxygen species caused by salt stress, while CAT decomposes hydrogen peroxide, thereby protecting the cells [57]. In this study, SOD significantly increased under salt stress (Figure 4A), which is consistent with previous research [58], while CAT levels significantly decreased under S treatment (Figure 4B), this is believed to be related to the serious growth inhibition in the salt stress treatment. Furthermore, SOD and CAT activity were significantly increased under G, SA and GSA treatments, which is consistent with previous studies [6,36]. Among these, the enzyme activity was highest under GSA treatment (Figure 4), likely due to the synergistic effect of grafting and exogenous SA combined treatment, resulting in an additive effect.”

Point 6: In the methodology, please explain the concept of what is a standard nutrient solution, see line 435?

Apologies for the confusion. We have revised the wording and indicated the EC and pH of the nutrient solution in line 521.

Point 7: A variety of solutions are used. Has the nutrient solution been modified in any way after the salt has been added so as not to alter its properties, especially its water potential?

Thank you. Regarding your question, I would like to clarify why NaCl was added to the nutrient solution for the salt stress experiment. While the addition of NaCl does indeed result in changes to the water potential of the nutrient solution, this change is a core component of the salt stress experiment. We seek to study how plants adapt to adverse conditions by altering water potential, as the presence of salt not only affects water potential but also significantly influences plant growth, root development, and physiological processes. This is the underlying rationale for our experimental design. And after added sodium chloride, the other nutrients in the nutrient solution remained unchanged and consistent.

Point 8: What is the origin of plants?

Jiaxina is a hybrid variety of tomato produced by Rijk Zwaan and purchased from Shandong Jinzhongzi Agricultural Development Co., Ltd. The rootstock Maxifort is a hybrid tomato variety produced by Bayer and purchased from Beijing Aoliwo Seed Technology Co., Ltd. We have listed the information in lines 488, 490.

Point 9: What was the light and temperature regime of the experiment? Was it natural light? This is the basis for setting parameters for measuring photosynthetic characteristics. Please clarify this.

During the entire period of salt stress treatment, the plants were placed in a modern greenhouse, where they were exposed to natural light conditions, and the day temperature about 24-28°C, night temperature about 16-18°C which described in lines 505-506. The photosynthetic rate was measured on clear days after 9 AM.

Point 10: It is necessary to check the literature review used. It is not cited uniformly. 

We have carefully reviewed the reference format according to the journal's requirements and added the missing DOIs for some references in lines 718, 753, 792, 810.

Reviewer 3 Report

Comments and Suggestions for Authors

The present version of “An enhanced interaction of graft and exogenous SA on photosynthesis, phytohormone and transcriptome analysis in tomato salinity stress” entitled manuscript required the below mentioned revisions.

1. In title, in tomato salinity stress is not complete phase, need complete meaning whether it is under or not.

2. Materials and methods section need drastic changes for making in details description about seedling growth, cultivation and treatment exposure. In which age, the salt stress was given is missing. Need to include the age of seedling for making graftage, while the complete procedure of grafting should be indicated. Why these two cultivars were used for grafting.  Is there any reason for using the Jiaxina' and 'Maxiford' as the scions and root stocks, respectively for grafting why not vice versa?  How many days salinity was given and in which days (after treatment exposure) data was taken. All these should be cleared in methodology.

3. Figure 10, The self-rooted seedlings of 'Maxiford' like Jiaxina should be added as treatment to compare among them and the effect of grafting and SA.

4. What about the salt-induced oxidative stress and consequence scenario in antioxidant defense in studied plants, it is recommended to add the oxidative stressor regulation by grafting and exogenous SA application as like osmotic and ionic stress.

Author Response

point 1. In title, in tomato salinity stress is not complete phase, need complete meaning whether it is under or not.

Thank you for your valuable feedback on our manuscript. Regarding your comment about the title, “in tomato salinity stress is not complete phase,” we would like to clarify that our experiments were indeed conducted under salinity stress conditions. Therefore, we propose to revise the title to: “An enhanced Interaction of Graft and Exogenous SA on Photosynthesis, Phytohormone, and Transcriptome Analysis in Tomato Under Salinity Stress”.

point 2. Materials and methods section need drastic changes for making in details description about seedling growth, cultivation and treatment exposure. In which age, the salt stress was given is missing. Need to include the age of seedling for making graftage, while the complete procedure of grafting should be indicated. Why these two cultivars were used for grafting.  Is there any reason for using the Jiaxina' and 'Maxifort' as the scions and root stocks, respectively for grafting why not vice versa?  How many days salinity was given and in which days (after treatment exposure) data was taken. All these should be cleared in methodology.

Thank you for your suggestions. We have revised the Materials and Methods section based on your feedback. We clarified the reasons for selecting Jiaxina and Maxifort as the scion and rootstock, respectively, in lines 488-492. The seedling cultivation method has been added in lines 492-494. Liness 494-495 now include the seedling age at the time of grafting, and lines 497-500 describe the detailed grafting procedure. The timing and duration of the salinity stress treatment is explained in lines 510 and 534, and the sampling times are specified in lines 534, 544, 576, 580, 586. We also added some additional details to make this section clearer.

point 3. Figure 10, The self-rooted seedlings of Maxifort like Jiaxina should be added as treatment to compare among them and the effect of grafting and SA.

Thank you for your valuable feedback. We appreciate your suggestion to include self-rooted seedlings of Maxifort for comparison. However, we believe this addition is unnecessary because Maxifort is a popular and widely used commercial rootstock variety, with a well-developed root system and good grafting compatibility. Many related studies, such as Interspecific Hybrid Rootstocks Improve Productivity of Tomato Grown under High-temperature Stress,  Modification of the Sensory Profile and Volatile Aroma Compounds of Tomato Fruits by the Scion × Rootstock Interactive Effect, Artificial Light for Improving Tomato Recovery Following Grafting: Transcriptome and Physiological Analyses, used Maxifort as rootstock to improve plant stress resistance and investigate its effects on scion growth, yield and fruit quality.

point 4. What about the salt-induced oxidative stress and consequence scenario in antioxidant defense in studied plants, it is recommended to add the oxidative stressor regulation by grafting and exogenous SA application as like osmotic and ionic stress.

Thank you for your constructive feedback. We have added the experiments on SOD and CAT activities, finished the figure and provided results in lines 168-178. Additionally, I have discussed the results in lines 396-408.

Round 2

Reviewer 2 Report

Comments and Suggestions for Authors

The authors submitted a revised version of the manuscript based on the reviewer's requests. The changes in the text have been noted and justified in the attached document. The changes have been accepted. After comparing the two versions of the manuscript, I can conclude that the artists have improved the quality of the manuscript and the manuscript has the prospect of being cited, so I recommend it for acceptance and publication.